# Risk factors for contracting malaria in six wards of Mudzi District, Zimbabwe: A case control-study

Tichaona Fambirai[1]*, Moses J. Chimbari[1], Pisirai Ndarukwa[1,2]

**1** School of Nursing and Public Health, College of Health Sciences, Howard College Campus, University of KwaZulu-Natal, Durban, South Africa, **2** Bindura University of Science Education, Faculty of Health Sciences, Bindura, Zimbabwe

* tfambirai@yahoo.com

## Abstract

Despite a significant decline in burden in the last two decades, malaria remains a significant global public health threat. Vector behaviour, climate, ecology, human economic and social behaviour, and quality of housing are some of the established predictors for contracting malaria. Zimbabwe has recorded a significant decline in malaria burden, however, districts like Mudzi continue to experience persistent malaria transmission despite well-performing indoor residual spraying programs. Persistent malaria transmission despite high IRS coverages point to human socio-economic behavior and health system factors, which have not been fully investigated. We therefore conducted this unmatched case-control study to identify human socio-economic, behavioural, and health system primary factors responsible for persistent malaria transmission in the district. We recruited 94 cases and 91 controls into the study. Cases were randomly recruited from health facility malaria treatment registers, whilst controls were neighbors of a case randomly recruited from a village household register. A case was defined as an individual residing in a selected village with a positive RDT test result. A control was an individual with no recorded positive RDT result recruited from the same village as a case. A structured questionnaire was used to collect data on socio-demographic characteristics of participants, behavior, perception of quality of health services, and knowledge on malaria causes, symptoms, and prevention. A key information guide was used to acquire perceptions of health managers on primary factors driving malaria in the district, as well as the performance of malaria control and treatment services. Kobo Collect was used for real-time data collection. Quantitative data were analyzed using STATA 13 (StataCorp LLC) to generate frequencies and odds ratios. Multivariate logistic regression analysis was conducted to identify independent risk factors for malaria. Independent risk factors for contracting malaria were: engaging in night outdoor social and religious activities (AOR = 8.13; 95% CI, 1.74–37.90), and having a garden (AOR = 4.51; 95% CI, 1.55–13.12). Wearing full body clothing at night (AOR = 0.13; 95% CI, 0.03–0.0.53),

**Data availability statement:** All relevant data are within the paper and its Supporting Information files.

**Funding:** The author(s) received no specific funding for this work.

**Competing interests:** The authors have declared that no competing interests exist.

and sleeping in a sprayed room (AOR = 0.04; 95% CI,0.01–0.31) were protective for contracting malaria. The majority of cases (96.74%) and controls (92.22%) had good knowledge of malaria transmission and preventative measures. Despite high knowledge, outdoor religious activities and outdoor socialization were significantly associated with contracting malaria. Increased night outdoor activity increases the likelihood of vector-human contact away from IRS-protected spaces. Sustaining IRS and intensifying integrated, targeted community engagement and malaria awareness programs will be key in eliminating malaria in Mudzi.

## Introduction

Despite global efforts to eliminate and eradicate malaria for over a century, malaria remains a public health threat, with over 200 million cases and approximately 450,000 deaths recorded annually [1]. Significant progress has been made in reducing the global malaria burden between 2000 and 2015. The reduction observed in the malaria burden has been underpinned by multiple interventions, chiefly indoor residual spraying, treatment with artemisinin combination therapies (ACTs), improved diagnostics, political commitment, and funding [2–4]. Despite this significant reduction in the malaria burden, approximately 50% of the global population is still at risk of contracting malaria ([5–7]. Gains made in malaria control in the last two decades face significant threats from climate change, vector and emerging malaria medicine resistance, emerging pandemics, and reduction in malaria control financial programmatic support and population movement [8,9]. Residual transmission [10,11] and border malaria [12] further exacerbate the complex malaria elimination drive. People living within malaria-endemic border regions face an ever-present threat of malaria risk due to persistent transmission and limited access to malaria interventions [12].

Pregnant women and children are at higher risk of contracting malaria compared to other population sub-groups [13–15]. Older populations and those infected with co-morbidities have also been shown to be at risk of contracting malaria [16,17]. Human population movement, insecticide resistance, drug resistance, vectoral changes [18], and human economic and social behaviors are significantly associated with the occurrence of malaria [19–21]. Adult populations engaged in outdoor occupations in agriculture, forestry, and mining have also been shown to be at a higher risk of contracting malaria [22–24]. Human outdoor activities, farming activities associated with irrigation, and outdoor and night activities have been documented as key drivers of malaria transmission in endemic settings and low-transmission zones [25]. Housing design and quality of materials used have also been shown to be significant predictors of malaria occurrence in communities [26,27]. Social and economic determinants have been shown to have a bearing on the level of risk of contracting malaria. Poverty has been positively associated with a higher risk for contracting malaria, as evidence has shown that low-income communities are at a significant risk for malaria compared to high-income communities [14]. Population movement has

also significantly contributed to malaria spread globally. In endemic border regions, movement across country borders has been associated with an elevated risk of contracting malaria [28–30].

Malaria transmission in Zimbabwe is seasonal and heterogeneous, with incidence positively correlated to the mean annual temperature range of 28–32˚C [31]. Furthermore, variability in malaria burden has been linked to peak rainfall periods, with the highest number of cases being recorded between November and April in the country [32]. Weekly malaria incidence for malaria often increases from an average of 100 cases during the dry season to over 1500 cases per week during the wet season in high malaria burdened district like Mudzi S1 Fig. Low-lying regions and middle-veld regions of the country record the highest number of cases compared to the highlands [31,33]. Malaria transmission in Zimbabwe is driven by *Anopheles gambiae* complex *(An. arabiensis ss* and *An. gambiae ss)* and *Anopheles funestus (An. funestus ss)* group vectors [34,35]. Recent national entomological surveillance data have revealed an early evening and early morning peak biting period and vectors with indoor and outdoor resting and biting capabilities [36].

Mudzi is among the top four (4) malaria-burdened districts in Zimbabwe with persistent malaria transmission despite a high-performing IRS program (>97% room and population coverages) within the past five (5) years [37] S2 Fig. Significant reduction in malaria burden has been observed in Zimbabwe since 2000 [32]. Despite the significant decline, malaria-endemic districts such as Mudzi continue to record high malaria incidence rates and Annual Parasite Index (API), ranging between 100 and 200/1000 population between 2016–2021 [32,38,39]. Entomology surveillance has shown the absence of insecticide resistance to IRS chemicals used in the national malaria vector control program [36,40]. Efficacy testing results have also been consistently within the WHO pesticide monitoring thresholds [36,40].

The primary factors for persistent malaria transmission in Mudzi, despite a high-performing IRS program, have not been fully explained. A broad project was conducted to evaluate the entomological, social, and health system primary factors responsible for persistent malaria transmission in Mudzi. Six sub-studies were conducted:

(i)   A study to identify vectoral composition, distribution, and abundance in six (6) wards.

(ii)  Focus Group Discussions to identify knowledge, attitudes, and perceptions among community members on malaria

(iii) Cross-sectional study incorporating parasitemia survey among artisanal miners and community members to identify practices, behaviour, and malaria positivity rates.

(iv) Cross-border movement assessment among malaria cases

(v)  An assessment of the health system to establish health system capacities to deal with malaria

The entomological studies sought to identify the vector behaviors, whilst this nested case-control study was specifically conducted to establish the human social and behavioral determinants of malaria in six (6) wards of Mudzi District. Understanding human practices and behaviour is key to developing appropriate interventions that minimize human-vector contact.

To the best of our knowledge, this is the first study conducted in multiple geographical zones within the district. A previous case study by Masango et al. was a reactive case-control study conducted to establish risk factors for malaria in one rural health center (RHC) that had recorded a malaria outbreak [33]. Zimbabwe's national malaria control program has set an ambitious goal to have at least twenty (20) districts with zero malaria transmission and to reduce districts conducting IRS programs from 31 to 23 by 2024 [32]. Our findings will contribute to malaria control and elimination efforts through evidence-based interventions in areas with persistent malaria transmission.

## Methods

This study was an unmatched case-control study with a case defined as a person with a positive RDT test result registered on the health facility malaria register between January and March 2023. A case was recruited from health facility registers, whilst controls were recruited from the same village as the case.

## Study setting

Zimbabwe is divided into ten (10) administrative provinces composed of 63 administrative districts. The districts are further subdivided into wards. Mudzi (Geo-coordinates: 17˚00′S032˚40′E) is one of the nine (9) administrative districts in Mashonaland East Province, located on the eastern border region of Zimbabwe and Mozambique, with a population of 165,266 [41]. Mudzi is a low agricultural potential area due to the low annual rainfall prevailing in the district (450–550 mm). Mudzi district is composed of eighteen [18] administrative wards. Each administrative ward is served by a single public health Rural Health Center (RHC), which offers an essential primary health care services package ranging from maternal child health, malaria treatment, immunization, and HIV and TB services.

**Sample size.** The sample size for the unmatched case-control study was calculated using the Fleiss formula in EPI Info ver 7.2.2® (Centers for Disease Control and Prevention) based on a study by Testahunegn et al.[42]. The assumption was that the proportion of cases with a history of night outdoor activity was 46.7% whilst the proportion of controls with a history of night outdoor activity was 14.7%, 80% power, and OR=3.7. The minimum sample size of 60 cases and 60 controls was calculated. Factoring in an attrition rate of 10% for both cases and controls, a total sample size was established at 66 cases and 66 controls.

## Sampling

**Study site.** This study was nested in a larger study that sought to investigate vectoral behavior, human knowledge and attitude, community practices, and health systems factors contributing to persistent malaria. The larger study employed a multi-stage sampling technique to select six (6) administrative wards out of the existing eighteen (18) wards. All eighteen (18) wards of Mudzi were grouped into six (6) clusters to ensure representativeness. The full description of the larger study's multi-stage sampling is fully described in S3 Fig.

A single ward was randomly selected from each of the six clusters using the random picker utility in Random List*. Six wards were selected for inclusion in the study (Ward 1, Ward 2, Ward 6, Ward 9, Ward 10, Ward 15), **Fig 1**. A single health facility was purposively selected for the study at each administrative ward.

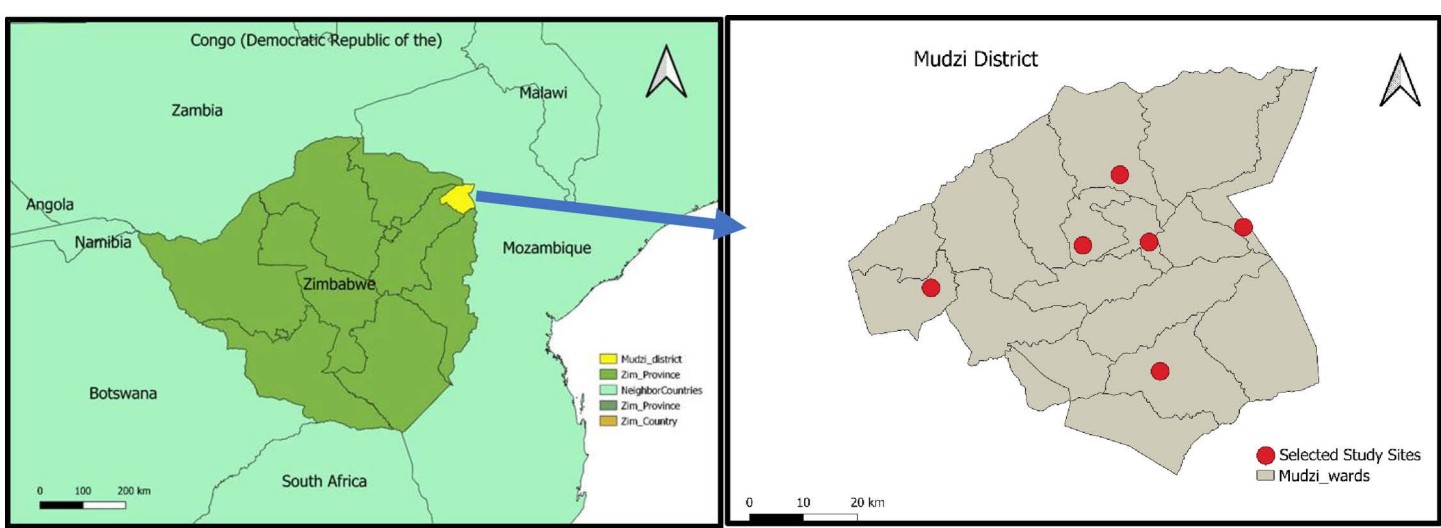

**Fig 1. Map of Zimbabwe and Mudzi District showing selected study sites.** * www.randomlist.com.

## Study participants

An equal proportion of cases from the calculated sample size was assigned to the six (6) selected wards, informed by similar malaria incidence rates. Cases were recruited from the health facility's malaria registers. The case serial numbers for cases diagnosed up to three months before the onset of the study were listed in the Microsoft Excel program and randomized. The random numbers were then sorted in increasing order. The number corresponding to the allocated sample size for each of the wards was recorded. The physical addresses of the cases were recorded for follow-up. Controls were recruited from the CHW village household list. The village household register was entered into a Microsoft Excel sheet, and a random listing was conducted. The random numbers were then sorted in increasing order, with the top corresponding numbers equating to the selected total cases for each village enrolled in the study. All the positive cases recorded in the health facility register and the CHW malaria treatment register were excluded from the control list.

## Eligibility

An individual living in the six (6) selected villages with an RDT-positive result for malaria recorded in the malaria facility register was selected for the study. An individual who tested positive for malaria and was not a resident of any of the six (6) villages was excluded from the study. Individuals who did not give written consent were excluded from the study.

## Data collection

A structured interviewer-administered questionnaire developed in English and translated into local language (*Shona)* was used to gather information on respondents' age, sex, occupation, level of education, occupation, outdoor activity, access to malaria health services, housing type, access to malaria prevention methods, previous history of malaria, and cross border movement. Information on the causes of malaria, malaria symptoms, and prevention methods was also solicited from cases and controls. Further information on the environments around the participants' households was collected through physical observations using the same tool. The questionnaire was developed and uploaded in English. The questionnaire was translated into *Shona* and back into English to check for consistency using back translation methods. The questionnaire was translated into *Shona* from English by two independent individuals blinded to the study's aims. The back-translated questionnaires were reviewed by all the researchers and research assistants (RAs) for consistency. All the RAs were trained on interviewing techniques. Pre-testing of the electronic data collection tools was conducted in one of the villages adjacent to a selected village. Ambiguous and difficult questions were reviewed and adjusted. Data were collected in real time using Kobo Collect® forms loaded on Android phones. The storage of collected data was done in Kobo Collect cloud servers. Data was collected over three weeks from 6–27 March 2023. Key informant interviews (KI) were also conducted to obtain additional perspectives on malaria transmission drivers, primary community socio-economic behaviour, and health system capacities with relation to access to treatment, surveillance, and services coverage. The KIs were also used to triangulate and validate responses from study participants. A structured key informant (KI) interview guide was used to interview malaria program managers namely the District Medical Officer (DMO, Districts Nursing Officer (DNO), District Environmental Health Officer (DEHO) District Laboratory Scientist (DLS), District Health Promotion Office (DHPO), Nurse in Charge Community (NIC), and District Pharmacy Manager (DPM). Cases and controls were interviewed at the homesteads, whilst KIs were interviewed at their respective workplaces.

## Data analysis

Data were downloaded from the Kobo Collect cloud storage platform as an Excel Spreadsheet and imported into Stata 13 for cleaning and analysis. Data was scanned for out-of-range values using scatter plots, frequency tables, and box and whisker plots. The Shapiro-Wilk test was used to test for normality on all continuous data variables. Normally distributed data were reported as mean (standard deviation), whereas non-normal data were reported as median (interquartile

range). Categorical data were presented as counts (frequency). Chi-square tests were used to test the association between categorical dependent and independent variables, as well as to identify significant differences in demographic variables between cases and controls. The Kruskal-Wallis test was used to identify significant differences in continuous demographic variables between cases and controls. Univariate analysis was used to identify independent variables with an effect on the outcome of interest (malaria). Crude odds ratios with 95% CI were calculated in the univariate analysis. Variables with a p-value of 0.25 or less from univariate analysis were put into the multivariable regression model to get the adjusted effect (AOR) of each independent variable on malaria. The higher cut-off p-value as a criteria for including variables in the multivariate model is based on the Hosmer and Lemeshow's purposeful selection of covariates, whereby in the univariate analysis, any variable having a significant univariate test at some arbitrary level (0.25 in our case) is selected as a candidate for the multivariate analysis [43]. This is based on the Wald test from logistic regression and a p-value cut-off of 0.25 [44]. More traditional levels, such as 0.05, can fail in identifying variables known to be important. An iterative stepwise forward multivariate logistic regression analysis was conducted to identify independent risk factors for malaria. Significance was set at a 95% confidence interval (CI). Qualitative data from key informants were manually sorted and analyzed by themes. A modified Bloom cut-off for total knowledge and attitude was used to assess knowledge of malaria using proportions scores using methods adopted in studies by Lopez et al.[45] and Okello et al. [46] with those with a score of 80–100% classified as good knowledge, 60–79-moderate knowledge, and <59%-poor knowledge. Maps showing study sites were generated using open-source QGIS software, utilizing shape files acquired online from the United Nations Office for the Coordination of Humanitarian Affairs (OCHA).

## Approvals and ethical considerations

Ethical approval was sought from the University of KwaZulu-Natal Bio-Medical Research Ethics Committee [BREC/00001594/2020] and the Medical Research Council of Zimbabwe [MRCZ/A/2637]. Gatekeeper permission was sought from the Provincial Medical Director (PMD) for Mashonaland East. Informed written consent was obtained from all the study participants before the administration of the questionnaire. Written parental/guardian consent was obtained for all study participants aged 18 years and below. Participants were informed of their rights and were free to discontinue participation at any time. Personal identifiers, such as full name, were only collected on consent forms; however, such information was excluded from analysis. Data was stored in a secure Kobo Collect cloud server with usernames and passwords granted only to the researchers. RAs' access to the Kobo Collect database was limited to the collection and submission of questionnaire data only. Submitted Kobo Collect forms from RAs were not retained on the individual phones' memory and were not shareable. Access for viewing collected data was only available to TF, PN, and MJC.

## Results

### Demographics

A total of 185 individuals were recruited into the study, 94 (50.81%) cases and 91(49.19%) controls, as shown in **Table 1**. The median (IQR) age of participants was 31 (19; 41.5) years.

Case and controls were comparable on demographic variables such as age, marital status, employment status, religion, and income distribution, with no statistically significant differences observed (p > 0.05), Table 1. The majority of cases and controls in this study had completed a secondary level education. Unemployment was highly prevalent among both cases and controls.

### Risk factors for malaria

In the univariate analysis, wearing clothes covering the whole body and engaging in night and early morning religious activities were significantly associated with contracting malaria, as shown in **Table 2**. In this study, those who slept in a

**Table 1. Socio-demographic characteristics of cases and controls in six wards of Mudzi District, Zimbabwe.**

| Variable | Cases n(%) | Controls n(%) | P-value |
|---|---|---|---|
| **Age** (Median, [IQR]) | 30.5 (18; 42) | 31 (22; 40) | 0.67 |
| **Gender** | | | |
| Male | 49(52.69) | 49(53.85) | 0.88 |
| Female | 44 (47.31) | 42 (46.15) | |
| **Age** | | | |
| 0–9 | 7(7.44) | 8(8.79) | 0.74 |
| 10–19 | 23(21.27) | 21(23.07) | |
| 20–29 | 20(24.46) | 22(24.17) | |
| 30–39 | 19(20.21) | 16(17.58) | |
| 40–49 | 13(13.82) | 10(10.98) | |
| 50+ | 12(12.76) | 14(15.38) | |
| **Marital status** | | | |
| Divorced | 3 (3.23) | 3 (3.30) | 0.67 |
| Married | 54 (58.06) | 57 (62.64) | |
| Minor | 13 (13.98) | 11 (12.09) | |
| Single | 22 (23.66) | 16 (17.58) | |
| Widow | 1 (1.08) | 4 (4.40) | |
| **Education Status** | | | |
| None | 3 (3.23) | 4 (4.44) | 0.87 |
| Primary | 19 (20.43) | 15 (16.67) | |
| Secondary | 68 (73.12) | 67 (74.44) | |
| Tertiary | 3 (3.23) | 4 (4.44) | |
| **Employment Status** | | | |
| Retired/pensioner | 2 (2.15) | 1 (1.10) | 0.75 |
| Self-employed | 14 (15.06) | 16 (17.58) | |
| Formally employed | 5 (5.38) | 6 (6.59) | |
| Student | 18 (19) | 13 (14.29) | |
| Unemployed | 52 (55.32) | 55 (60.44) | |
| **Religion** | | | |
| African Apostolic | 31 (33.33) | 22(24.18) | 0.39 |
| African Traditional | 16 (17.20) | 15 (16.48) | |
| Catholic | 5 (5.38) | 2 (2.20) | |
| None | 27 (29.03) | 39 (42.86) | |
| Pentecostal | 11 (11.83) | 11(12.09) | |
| Protestant | 3 (3.23) | 2 (2.20) | |
| **Estimated monthly income** (Median [IQR]) (US$) | 21.50(5.00; 47.50) | 20.00(10.00; 50.00) | 0.59 |

room sprayed with IRS chemical and wore clothes covering the whole body during night outdoor activities were less likely to contract malaria. In this study, those who reported a history of cross-border movement, cooking outside at night, engaging in artisanal mining, and living in a house with open eaves were not significantly associated with contracting malaria.

Potential malaria confounders such as age, house status, income, access to a community health worker, and treated net ownership were accounted for in a multivariate logistic regression model fitted onto the data using a forward stepwise method. Attending religious services at night and morning, socializing outside at night, and owning a garden were

**Table 2. Univariate analysis of potential risk factors for contracting malaria in six wards of Mudzi district, Zimbabwe.**

| Variable | Case (n=94) | Control (n=91) | Odds Ratio (OR) | 95% CI | P-value |
|---|---|---|---|---|---|
| **Agricultural Activities** | | | | | |
| Own a garden | | | | | |
| Yes | 36 | 25 | 1.49 (ref) | 0.80; 2.80 | 0.18 |
| No | 38 | 40 | | | |
| Engage in morning field/gardening activities | | | | | |
| Yes | 40 | 36 | 1.45(ref) | 0.78; 2.71 | 0.21 |
| No | 36 | 29 | | | |
| Return from the fields/garden after sunset | | | | | |
| Yes | 40 | 30 | 1.70(ref) | 0.90; 3.23 | 0.08 |
| No | 35 | 35 | | | |
| **Personal protection** | | | | | |
| Put on clothes covering the whole body at night | | | | | |
| Yes | 22 | 35 | 0.44(ref) | 0.23; 0.87 | 0.01 |
| No | 53 | 31 | | | |
| **Indoor residual spraying Status** | | | | | |
| Homestead was sprayed last year | | | | | |
| Yes | 57 | 72 | 0.40(ref) | 0.19; 0.81 | 0.01 |
| No | 36 | 18 | | | |
| Sleep in a sprayed room | | | | | |
| Yes | 13 | 20 | 0.24(ref) | 0.06; 0.81 | 0.01 |
| No | 60 | 44 | | | |
| **Treated Nets Utilisation** | | | | | |
| Own a treated mosquito net | | | | | |
| Yes | 21 | 70 | 0.55(ref) | 0.27; 1.12 | 0.07 |
| No | 31 | 57 | | | |
| Sleep under a treated mosquito net | | | | | |
| Yes | 18 | 44 | 0.67(ref) | 0.31; 1.44 | 0.27 |
| No | 31 | 51 | | | |
| Use mosquito repellents in the household | | | | | |
| Yes | 9 | 13 | 0.65(ref) | 0.30; 1.40 | 0.23 |
| No | 66 | 52 | | | |
| **Outdoor Activity** | | | | | |
| Engage in night and morning outdoor religious services | | | | | |
| Yes | 25 | 9 | 4.47(ref) | 2.00; 10.52 | <0.001 |
| No | 49 | 57 | | | |
| Engage in artisanal mining | | | | | |
| Yes | 4 | 4 | 1.38(ref) | 0.36; 5.75 | 0.59 |
| No | 70 | 62 | | | |
| Socialize outside at night | | | | | |
| Yes | 39 | 17 | 2.92(ref) | 1.51; 5.67 | <0.001 |
| No | 35 | 48 | | | |
| Cook outside in the evening | | | | | |
| Yes | 22 | 17 | 1.51(ref) | 0.76; 3.01 | 0.21 |
| No | 53 | 49 | | | |
| Perform outdoor household chores early morning or evening | | | | | |
| Yes | 41 | 41 | 0.97(ref) | 0.51; 1.86 | 0.92 |

*(Continued)*

**Table 2.** (Continued)

| Variable | Case (n=94) | Control (n=91) | Odds Ratio (OR) | 95% CI | P-value |
|---|---|---|---|---|---|
| No | 32 | 25 | | | |
| **Habitation Status** | | | | | |
| Visible breeding sites around homestead | | | | | |
| Yes | 14 | 8 | 1.82(ref) | 0.62-5.27 | 0.20 |
| No | 79 | 82 | | | |
| Building materials (Pole and Dagga) | | | | | |
| Yes | 6 | 7 | 0.82(ref) | 0.22-2.98 | 0.73 |
| No | 87 | 83 | | | |
| Building materials (brick and mortar) | | | | | |
| Yes | 46 | 51 | 0.75 (ref) | 0.30-1.40 | 0.30 |
| No | 47 | 31 | | | |
| Open Eaves | | | | | |
| Yes | 23 | 15 | 1.64 (ref) | 0.75-3.67 | 0.18 |
| No | 70 | 75 | | | |
| **Cross-Border Movement** | | | | | |
| Cross the border into Mozambique | | | | | |
| Yes | 41 | 41 | 1.4(ref) | 0.37; 5.81 | 0.58 |
| No | 32 | 25 | | | |

independently associated with an increased risk of contracting malaria **Table 3**. Sleeping in a sprayed room and putting on clothes covering the whole body reduced the odds of contracting malaria by 96% and 87%, respectively.

## Knowledge of malaria transmission

Knowledge of malaria transmission was high among cases and controls, with 96.74% of cases and 92.22% of controls being aware of what causes malaria. General body weakness (52.13% of cases and 48.35% of controls) and loss of appetite (11.7% of cases and 13.19% of controls) were the two most reported signs and symptoms of malaria identified, as depicted in **Table 4**. Significant differences in knowledge between cases and controls were noted in malaria prevention actions ($p < 0.05$) and knowledge of the effectiveness of the chemical used to spray ($p < 0.001$).

## Health service delivery

There was no significant difference between the views on quality of service at health facilities between cases and controls ($p > 0.05$). The majority of cases (71.11%) and controls (70.00%) were of the view that the service offered at health facilities was good **Table 5**.

**Table 3. Multivariate logistic regression analysis of risk factors for contracting malaria in Mudzi, Zimbabwe.**

| Variable | AOR | 95% CI | P-value |
|---|---|---|---|
| Sleep in a sprayed room | 0.04 | 0.01; 0.31 | <0.001 |
| Put on clothes covering the whole body when going out at night | 0.13 | 0.03; 0.53 | <0.001 |
| Socialize outside at night | 2.94 | 1.09; 7.96 | 0.03 |
| Engages in night and morning outdoor religious activities. | 8.13 | 1.74; 37.90 | <0.001 |
| Own a garden | 4.51 | 1.55; 13.12 | <0.001 |

**Table 4. Reported knowledge of malaria transmission and prevention among cases and controls in Mudzi district, Zimbabwe.**

| Variable | Cases (n=92) n (%) | Controls (n=90) n(%) | P-value |
|---|---|---|---|
| Know cause of malaria | | | |
| Yes | 89(96.74) | 83(92.22) | 0.18 |
| No | 3(3.26) | 7(7.78) | |
| Knowledge of contracting malaria | | | |
| Mosquito bite | 78 (83.87) | 74 (82.22) | |
| Eating infected mangoes | 5 (5.38) | 3 (3.33) | |
| Flies | 6 (6.45) | 6 (6.67) | |
| No knowledge | 3 (3.23) | 7 (7.78) | 0.53 |
| Evil spirits | 1 (1.08) | – | |
| Signs and symptoms of malaria | | | |
| Headache | 3 (3.19) | 1 (1.10) | 0.44 |
| General body weakness | 8 (8.51) | 16 (17.58) | |
| Vomiting | 12 (12.77) | 8 (8.79) | |
| Pain in joints | 11 (11.70) | 10 (10.99) | |
| Loss of appetite | 11(11.70) | 12 (13.19) | |
| Fever | 49 (52.13) | 44 (48.35) | |
| Malaria prevention actions known | | | |
| Cutting grass around the homestead | 2 (2.17) | 3 (3.33) | 0.05 |
| Use a mosquito net | 11 (11.96) | 6 (6.67) | |
| Spraying homes | 65 (70.65) | 57 (63.33) | |
| Wearing long clothes at night | 10 (10.87) | 23 (25.56) | |
| No knowledge | 4 (4.35) | 1 (1.11) | |
| Perception of the efficacy of IRS chemicals | | | |
| Perceive IRS chemical as effective | | | |
| Yes | 51 (56.04) | 73 (83.91) | <0.001 |
| No | 9 (9.89) | 5 (5.75) | |
| Not sure | 31 (34.07) | 9 (10.34) | |

**Table 5. Perspectives on quality of malaria treatment service among cases and controls.**

| Variable | Cases (n=76) n(%) | Controls (n=66) n(%) | P-value |
|---|---|---|---|
| Poor | 2(2.22) | 2(2.50) | 0.45 |
| Good | 64(71.11) | 56(70.00) | |
| Fair | 12(13.33) | 11(13.75) | |
| Excellent | 10(11.11) | 5(6.25) | |
| No comment | 2(2.22) | 6(7.50) | |
| Bad | 0(0) | 0(0) | |

## Malaria managers' perspectives on malaria transmission and programmatic capacities and coverages

A total of seven (7) district health team (DHT) managers were interviewed for their technical opinions and perceptions on malaria program implementation and transmission. DHT managers reported that only 359/ 503 (71%) villages have a community health worker, leaving a sizeable proportion uncovered. The majority of the local population in the selected

villages is engaged in artisanal mining along the major rivers within the district. The miners are exposed to outdoor bites as they will be working and living in the open.

The DMO reported that:

*"Artisanal mining is a major source of livelihood in this community due to the area's low agricultural potential. Therefore, the majority of unemployed people in the community engage in artisanal gold mining for survival."*

Night social activities at service centers and religious services were reported by the majority of the managers as key drivers of local malaria transmission. The DEHO reported:

*"The majority of the local male population frequent local shops in the evening for socialization purposes and entertainment."*

*"There is a large proportion of the local population who attend religious meetings and worship services during early morning and evenings. Sometimes, the events last for days whilst the gatherers are not protected from malaria. At these events, seldom do people protect themselves from mosquito bites."*

The health managers reported that communities in Wards 1 and 2 practice shifting cultivation. During the farming season (November-April), the majority of the community members reside at their farming plots far away from their sprayed homesteads. The district is currently not conducting routine LLINS/ITNs distribution to complement IRS program. The DNO reported that at least ninety percent (90%) of the nursing staff and CHWs have received training in malaria case management. On the other hand, only a minority of HCWs have received full training on Integrated Disease Surveillance and Response (IDSR). The uncontrolled cross-border movement of the local population into Mozambique was thought of by the managers as a key driver for malaria transmission in the district.

*"Local communities have historical, social, and economic ties with communities across the border in Mozambique. Uncontrolled transboundary movement is a common feature here. Whilst there is an ongoing IRS program here in Zimbabwe, in the adjacent Mozambique district, there are no routine IRS activities. "*

Whilst the district runs routine social behaviour change and communication (SBCC) and pre-IRS awareness and community mobilization programs, these efforts are inadequate to reach the majority of the population. CHWs have not been effectively disseminating routine health education messages on malaria due to limited transport means. The DHPO reported:

*"It would be good if there were adequate transport and financial means to conduct routine community malaria SBCC and community engagement activities".*

The NIC added*:*

*"The majority of our village health workers (VHWs) do not have bicycles, hence their activities are mostly limited to households within a walking distance from their place of residence."*

## Discussion

Human socio-economic behaviour plays a critical role in sustaining malaria transmission. This study was conducted to establish human behaviour determinants for persistent transmission despite an effective IRS program in six (6) rural

wards of the Mudzi district. Vector control interventions were protective, whilst night outdoor activities were significantly associated with contracting malaria. Both cases and controls had good knowledge of malaria; however, misconceptions were observed. Health systems challenges impede the delivery of community-based malaria control activities.

This study confirmed the established protective effect of IRS, which has been proven globally [47,48]. Additionally, the use of LLINS/ITNs has contributed to the decline in global malaria burden [49–51]. Contrary to other studies [52,53], our study did not establish the protective effect of treated bed net use. Correct use, quality of the net, and LLINSs/ITNs' chemical efficacy were not assessed in this study; however, these factors could likely explain the lack of protective effect. Studies conducted elsewhere have attributed the absence of protective effect of LLINS/ITNs to net structural damage and over-washing at short intervals [54–56] as well as bed net sharing [55]. Dual use of LLINS/ITN and IRS has shown both beneficial synergistic outcomes attributed to dual IRS and ITN intervention compared to IRS alone [57,58]. Despite the benefits, it is also critical to consider the antagonistic effect of the intervention resulting from dual intervention [57]. Encouraging the use of self-acquired and quality-assured treated nets whilst ensuring timely replacement and maintenance of treated nets may offer significant additional protection against malaria in the district.

Social and religious influences on nighttime outdoor movement increase exposure to mosquito bites. Night and morning outdoor movements have been established as primary factors for contracting malaria [59–61]. Community members engaging in night outdoor activities without any protective measures in the district is highly probable, as findings from other studies in similar settings have shown individuals engaging in night social and economic activities without any protective measures [62,63]. Mudzi experiences higher temperatures throughout the year. Elevated temperatures obtained in the district, coupled with poorly ventilated housing structures, create a warmer indoor environment, which forces people to stay outdoors at night [33]. Similar findings were observed in Tanzania [63], where warmer indoor conditions were responsible for individuals staying outdoors at night, exposing themselves to outdoor mosquito bites. Wearing long clothes covering the whole body is a viable, low-cost, and effective barrier against mosquito bites for the local population, as it has proved protective in this study and elsewhere [33,64]. Promoting community members to wear long clothes when outdoors at night should be reinforced through community health education initiatives and awareness programs. Robust risk behavior change strategies should be implemented to reduce unprotected night outdoor activities to effectively reduce malaria transmission [29,42–45].

Despite the lack of significant association between the occurrence of malaria and cross-border movement in quantitative analysis, KIs cited this activity as a key driver for malaria in the district. Contrary to our findings, other studies have established a significant association between cross-border movement and contracting malaria in border regions [65–67]. Cross-border movement poses a significant threat to malaria control efforts as Individuals moving across borders with limited access to malaria protection interventions [68–70]. The complex nature of human population movement poses challenges in developing appropriate interventions for mobile populations [71]. Increasing health education at points of entry and malaria screening has been recommended as a viable mitigatory measure [72].

KIs implicated staying at crop fields away from IRS-sprayed homes as a possible driver of malaria within the district. Migratory farming has been associated with exposure to elevated night outdoor activity coupled with poor shelter [73]. Long outdoor exposure coinciding with peak biting periods and the presence of efficient malaria vectors within the district increases the risk of contracting malaria. Routine vector surveillance data in the district have shown high abundance and distribution of *An. gambiae sl,* and *An. funestus sl* with mixed outdoor and indoor resting and feeding capabilities [36,40]. Inhibiting human nighttime outdoor practices that coincide with peak biting phases is vital. Gardening is a source of livelihood for the majority of the local population; however, in this study, engaging in gardening activities was significantly associated with contracting malaria. Garden farming has the potential to create favorable habitats essential for mosquito breeding [74,75], which increases the likelihood of human-vector contact and mosquito abundance [47,49]. Our study findings are comparable to other studies, which reported a significant association between farming activities and contracting malaria [76,77]. Environmental control measures such as larviciding of breeding spaces within the garden fields could be a viable option that may result in minimal disruption of farming processes whilst reducing mosquito abundance [78].

The high knowledge obtained in this study could be attributed to higher literacy levels among the study participants, as higher education levels have been associated with elevated knowledge levels [79,80]. Knowledge on malaria has been shown to be higher in malaria endemic regions as a result of routine exposure to effective social behaviour change and communication (SBCC) programs [81,82]. A comparable level of high knowledge was also observed in similar studies in Malaysia by Rahim et al. [60] and in Ghana by Adum et al.[83]. Despite this high knowledge, significant risk for contracting malaria exists in the district, likely due to social and economic activities. Whilst this study revealed a non-significant association of engaging in artisanal gold mining (AGM) due to the low proportion of participants engaging in the activity, KIs implicate AGM as a key malaria driver in the district. AGM is significantly associated with the occurrence of malaria in various settings [84–86] due to its association with limited access to health care, abundant mosquito breeding sites, and prolonged outdoor living. AGM and night religious activities reduce the effectiveness of IRS protection as individuals spend time outdoors during mosquito peak biting periods. The use of personal topical repellants and spatial repellents is a viable option to minimize outdoor mosquito bites; however, the cost has been established to be prohibitive for low-income communities [87,88]. Increasing access to health education and CHWs outreach program at AGM sites may increase awareness and access to care. This calls for novel integrated, targeted community engagement and social behaviour intervention focused on at-risk groups.

Significant differences in knowledge of preventative measures and the perceived low efficacy of IRS chemicals between cases and controls could lead to differential uptake of the two interventions in the community. Perceived low efficacy of IRS chemicals among communities has been reported in Ghana [89], where participants viewed modern IRS chemicals as weaker compared to previously utilised IRS chemicals. Providing correct education on IRS chemical efficacy during IRS campaigns is critical in improving the perceived potency of IRS chemicals as well as uptake of the intervention [89].

The positive perceived quality of health care service received at health facilities will facilitate the utilisation of standard malaria diagnosis and treatment. This preference for orthodox health services is threatened by a myriad of challenges, including inadequate human resources, financial resources, and limited transport coverage for health care workers and CHWs. The non-presence of CHWs in some villages and inadequate transport to increase coverage result in limited access to malaria treatment and health education. Increasing CHW coverage has been linked to improved access to treatment coverage at the community level in Liberia [90]. Evidence from Kenya shows that CHWs are highly regarded in their local communities and their community-based services reduce distance to health facilities as well as waiting time [91]. Increasing investment in health system strengthening through CHW recruitment, training, and adequate financing of community malaria awareness programs will be key in reducing the malaria burden in the district.

This study had limitations. Recall bias in this study cannot be ruled out as cases were interviewed after recovery and, in some instances, up to three months after illness. Cases could likely have recalled their exposure experiences more than controls. The use of self-reported control measures, night outdoor activity, and treated net usage without observation limits the generalization of the results. Recruitment of control based on absence and non-appearance in treatment registers may have introduced misclassification bias as some individuals could have sought treatment outside the district or selected study wards. The misclassification bias could likely have elevated or suppressed the effect observed in the study. Despite these limitations, valuable inferences can be made relevant to the study area. This study is the first attempt to establish socio-economic behavioural factors and health systems factors contributing to persistent malaria transmission in Mudzi district.

## Conclusion

This study confirms the significant role of night outdoor activities as a significant determinant of the occurrence of malaria transmission in Mudzi. Elevated night outdoor activity in a region with vectors with outdoor feeding and resting capabilities increases human-vector contact outside IRS-protected living spaces. Reducing malaria burden in this district will require

investment in complementary interventions that influence human socio-economic behavior. Integrated targeted social behavior change programs for key groups engaged in migratory farming, artisanal mining, and religious activities may have a significant impact. Promoting low-cost interventions such as wearing long clothes that cover the whole body when engaged in night-outdoor activities should be promoted. There is a need to strengthen continuous community engagement programs on malaria prevention to translate knowledge into actual preventative behaviour. Continuous investment in health system strengthening through expansion of the community health workforce, adequate financing of behaviour change programs, tooling CHWs, and enhancing cross-border malaria control collaborative initiatives will be critical in reducing malaria burden in Mudzi district.

## Supporting information

**S1 Fig. Weekly Malaria trends in Mudzi District: 2015–2021.**
(TIF)

**S2 Fig. Indoor residual coverage rates and malaria incidence rate: 2017–2023, Mudzi District, Zimbabwe.**
(TIF)

**S3 Fig. Multi-stage sampling technique for malaria study in Mudzi, Zimbabwe.**
(TIF)

## Author contributions

**Conceptualization:** Tichaona Fambirai, Pisirai Ndarukwa.

**Data curation:** Tichaona Fambirai.

**Formal analysis:** Tichaona Fambirai.

**Investigation:** Tichaona Fambirai.

**Methodology:** Tichaona Fambirai, Moses J. Chimbari.

**Supervision:** Moses J. Chimbari, Pisirai Ndarukwa.

**Writing – original draft:** Tichaona Fambirai.

**Writing – review & editing:** Tichaona Fambirai, Moses J. Chimbari, Pisirai Ndarukwa.

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
