## [Decision Letter · Decision Letter 0]

PONE-D-24-44171Risk Factors for Contracting Malaria in Six Wards of Mudzi District, Zimbabwe: A Case Control-Study.PLOS ONE

Dear Dr. Fambirai,

Thank you for submitting your manuscript to PLOS ONE. After careful consideration, we feel that it has merit but does not fully meet PLOS ONE’s publication criteria as it currently stands. Therefore, we invite you to submit a revised version of the manuscript that addresses the points raised during the review process.

We look forward to receiving your revised manuscript.

Kind regards,

Rajib Chowdhury, M.Sc.; MPH

Academic Editor

PLOS ONE

Journal Requirements:

3. We note that you have referenced (Unpublished) on page 25, which has currently not yet been accepted for publication. Please remove this from your References and amend this to state in the body of your manuscript: (ie “Bewick et al. [Unpublished]”) as detailed online in our guide for authors

4. We note that [Figure 1] in your submission contain [map/satellite] images which may be copyrighted. All PLOS content is published under the Creative Commons Attribution License (CC BY 4.0), which means that the manuscript, images, and Supporting Information files will be freely available online, and any third party is permitted to access, download, copy, distribute, and use these materials in any way, even commercially, with proper attribution. For these reasons, we cannot publish previously copyrighted maps or satellite images created using proprietary data, such as Google software (Google Maps, Street View, and Earth). For more information, see our copyright guidelines: http://journals.plos.org/plosone/s/licenses-and-copyright.

Reviewers' comments:

Reviewer's Responses to Questions

**Comments to the Author**

1. Is the manuscript technically sound, and do the data support the conclusions?

Reviewer #1: Yes

Reviewer #2: Yes

Reviewer #3: Yes

2. Has the statistical analysis been performed appropriately and rigorously? 

Reviewer #1: Yes

Reviewer #2: Yes

Reviewer #3: Yes

3. Have the authors made all data underlying the findings in their manuscript fully available?

Reviewer #1: Yes

Reviewer #2: Yes

Reviewer #3: Yes

4. Is the manuscript presented in an intelligible fashion and written in standard English?

Reviewer #1: Yes

Reviewer #2: Yes

Reviewer #3: Yes

5. Review Comments to the Author

Reviewer #1: Reviewer Recommendation and Comments for Manuscript Number PONE-D-24-44171________________________________________

Major Comments

1. Research Gap and Study Objective (Lines 11–17):

o The abstract begins with the global impact of malaria and provides context for the study's importance. However, despite the well-performing IRS, the research team highlights the persistence of malaria transmission in Mudzi. For example:

"Despite significant reductions in the malaria burden in Zimbabwe, persistent high transmission in districts like Mudzi indicates a potential role of behavioral and environmental factors unaddressed by current interventions."

o Clearly articulate how the study addresses a novel aspect or adds to existing literature.

2. Study Design and Methods (Lines 17–23):

o The methods are well-described, but the recruitment process for cases and controls needs clarification. Specifically:

Were controls matched in any way beyond being neighbours? If not, why was an unmatched design chosen?

Specify the criteria for classifying a participant as a "case" beyond malaria facility registers.

o The use of the Kobo Collect tool is noted but lacks justification. Add a brief rationale for its use (e.g., efficiency, accuracy, or real-time data collection in rural settings).

3. Key Findings and Interpretation (Lines 24–28):

o The findings are clearly reported with adjusted odds ratios (AOR) and confidence intervals. However:

The abstract lists the results but does not provide sufficient interpretation. For example, why might outdoor night activities (religious and social) and gardening increase risk? How do these findings align with or differ from other studies?

Briefly address why knowledge about malaria (high in both groups) did not translate to reduced risk for cases.

4. Conclusions and Recommendations (Lines 31–33):

o The conclusion emphasizes behavior change strategies to complement IRS but lacks specific recommendations. Strengthen this by suggesting actionable measures, such as targeted educational campaigns or community-based interventions during high-risk periods (e.g., night religious gatherings).

Major Comments

1. Clarity and Presentation of Demographics (Lines 193–201):

o The demographic results mentioned are precise but could be more concise and reader-friendly. For example, instead of stating, "There were no significant differences (p>0.05) between cases and controls on demographic variables," consider rephrasing to group insights, such as:

"Age, marital status, employment status, and income distribution were comparable between cases and controls, with no statistically significant differences observed (p>0.05)."

o Ensure the comparisons align with the data shown in Table 1 and highlight any relevant demographic findings.

2. Sampling and Study Design (Lines 122–139):

o The explanation of sampling is detailed but lacks clarity in connecting the rationale for multi-stage sampling to the study objectives. Consider explicitly justifying the choice of six wards and how they adequately represent the district.

o It might help to explain why neighbours of cases were chosen as controls and how potential confounding factors, such as shared environmental risks, were accounted for.

3. Key Informant Interviews (Lines 157–164):

o The role of the key informant interviews in complementing the case-control study is not clearly explained. What specific insights or triangulation did the interviews add to the findings? A clear link between these data and the overall research objectives would enhance the manuscript.

4. Data Analysis (Lines 166–179):

o The criteria for selecting variables with a p-value of 0.25 for logistic regression should be briefly justified, as this threshold may not be intuitive to all readers.

o The Shapiro-Wilk test is used for normality testing, but it would help to mention whether data transformations were applied for non-normal data or how this affected subsequent analyses.

5. Knowledge Scoring (Lines 180–181):

o The method for assessing malaria knowledge is straightforward but would benefit from additional context. How were these knowledge thresholds determined? Were they adapted from a prior study or guideline? Clarify how knowledge was assessed across the case and control groups.

6. Ethics and Approvals (Lines 183–190):

o Researchers ensured participant confidentiality during the physical follow-ups in villages, but the methods used to maintain it are not mentioned in the ethical considerations. Adding a brief note on this aspect would strengthen the ethical rigor of the manuscript.

Major Comments

1. Clarity and Presentation of Demographics:

o The authors can improve the narrative describing the findings (lines 193–201) to enhance the demographic results' readability in Table 1. For example, the statement "There were no significant differences (p>0.05) between cases and controls on demographic variables" could be supported with more concise explanations or grouped insights, such as by stating, "Age, marital status, employment status, and income distribution were comparable between cases and controls."

2. Risk Factors:

o The analysis of risk factors (lines 204–219) is valuable, but there is some redundancy in presenting odds ratios in both text and tables. Consider summarizing only the most significant findings in the text (e.g., wearing protective clothing, sleeping in sprayed rooms, and attending church services) while directing readers to the table for full details.

o The multivariate analysis (Table 3) is compelling, but the forward stepwise method used for model building lacks detail. Explain the criteria used to include or exclude variables in the model, as this is critical to assess the robustness of the findings.

3. Knowledge of Malaria Transmission:

o The authors could further discuss the statistical significance of differences in prevention actions (p=0.05) and perceptions of chemical efficacy (p<0.001) to explore their practical implications for public health interventions . For example, how can this information be leveraged to enhance malaria prevention strategies?

4. Health Service Delivery:

o The section on health service delivery (lines 231–233) is underdeveloped. While it is stated that most participants felt they received good treatment, the specific measures used to assess "good treatment" or "friendly staff" are unclear. Elaborate on how these perceptions were measured (e.g., through structured interviews or surveys) and consider the potential for recall bias in self-reported data.

5. Statistical Reporting and Interpretation:

o The presentation of p-values and confidence intervals is appropriate; however, the manuscript lacks a discussion on potential confounding factors that may affect the observed associations. For example, were socioeconomic factors or healthcare access considered potential confounders?

o In Table 3, the AOR for attending morning and night church services (8.13; 95% CI: 1.74–37.90) has a wide confidence interval, suggesting limited precision. This warrants further discussion regarding potential issues such as sample size or variability in this subgroup.

6. Ethical Considerations and Study Limitations:

o There is no mention of ethical approval or participant consent in the provided text. Ensure this information is clearly stated in the manuscript.

o The authors should acknowledge the study's limitations, including reliance on self-reported behaviors and potential selection bias in recruiting cases and controls. They should also discuss the generalizability of the findings to other populations or regions 

Minor Comments

1. Terminology Consistency (Lines 15–17):

o Replace "well-performing indoor residual spraying program" with more precise language, such as "effective indoor residual spraying (IRS) program," to avoid redundancy.

2. Statistical Analysis Details (Lines 23–24):

o Replace "stepwise forward multivariate logistic regression analysis" with a more concise phrase, such as "multivariable logistic regression analysis." Avoid overly technical jargon in the abstract.

3. Protective Factors (Lines 27–28):

o The phrase "putting on long clothes that cover the whole body at night" is wordy. Simplify to "wearing full-body clothing at night."

4. Knowledge Assessment Results (Lines 28–30):

o The phrase "knew at least one major method of controlling malaria" is vague. Specify what constitutes a "major method" (e.g., use of mosquito nets, IRS).

5. Key Drivers of Malaria Transmission (Lines 31–32):

o Replace "key drivers of malaria transmission" with "primary risk factors for malaria transmission" to align with scientific terminology.

6. Keywords (Lines 34):

o The keywords are relevant but could be expanded to include "Indoor Residual Spraying," "Behavior Change," and "Case-Control Study" to improve discoverability.

Minor Comments

1. Terminology Consistency (Lines 93–95):

o The manuscript switches between "RDT test" and "positive RDT test result." For consistency, use a single term like "positive RDT result."

2. Geographic Description (Lines 98–104):

o The description of Mudzi district could benefit from a more concise summary. For example, "Mudzi, located on the Zimbabwe-Mozambique border, has 21 administrative wards serving a population of approximately 165,266 (2022 Census)."

3. Use of Abbreviations (Lines 104–106):

o Avoid overloading the reader with abbreviations in a single sentence (e.g., RMNCH, EPI, HIV & TB). Consider explaining only the most relevant terms in the main text and moving others to a footnote or supplementary material.

4. Tool Translation (Lines 155–156):

o While the translation of tools into Shona is commendable, you should specify how translation accuracy and consistency were ensured (e.g., through back-translation or pre-testing).

5. Sampling Tool Reference (Lines 126–127):

o The inclusion of "www.randomlist.com" in the text might be better placed in a footnote or omitted to maintain a formal tone.

6. Figure 1 Placement (Lines 99–129):

o The reference to Figure 1 is ambiguous. Ensure the figure is placed appropriately within the manuscript and fully described in the caption.

7. Data Cleaning Description (Lines 167–169):

o While you describe data cleaning methods like scatter plots and frequency tables, it may help to briefly state what common outliers were found (if any) and how they were handled.

8. Typographical Issues:

o Line 104: Replace "25 )(39)" with a clearer reference format.

o Line 119: Clarify whether the total sample size was 132 or 66 cases and 66 controls. Avoid redundancy.

Minor Comments

1. Grammar and Syntax:

o Lines 194–195: Replace "94 [50.81%] cases and 91[49.19%]" with "94 (50.81%) cases and 91 (49.19%) controls" for consistency.

o Line 223: "General body weakness (cases – 52.13%, controls – 48.35%) and loss of appetite..." could be simplified to "General body weakness (52.13% of cases and 48.35% of controls) and loss of appetite..."

2. Table Presentation:

o In Table 1, ensure consistent formatting for percentages and alignment of columns. For instance, percentages should be rounded to one or two decimal places for clarity (e.g., 73.12% and 74.44%).

o Tables 2 and 3 should include clear captions explaining the analysed variables, methods, and statistical tests.

3. Terminology:

o Avoid ambiguous terms such as "good treatment" (line 231). Specify criteria or metrics used for this assessment.

o Replace "the odds of suffering from malaria" (line 216) with "the odds of contracting malaria" for accuracy.

4. Formatting:

o Ensure tables and figures follow PLoS ONE formatting guidelines, including appropriate captions and footnotes for statistical results.

o The text should use a consistent style for referencing p-values (e.g., p = 0.05 instead of p=0.05).

Reporting Guidelines (STROBE)

Following the STROBE guidelines (Strengthening the Reporting of Observational Studies in Epidemiology) is essential for observational studies like this. The manuscript does a good job of detailing the study design, sampling, and data collection procedures. However, make sure that all relevant STROBE items are addressed, including:

• Participant flow: Were any cases or controls excluded after enrollment? This should be clearly stated.

• Confounding variables: Were potential confounders measured, and how were they controlled for in analysis?

• Outcome measures: Clearly define the primary and secondary outcome measures and describe how they were measured.

Explicitly mention participant flow, including the number of cases/controls excluded and why, and confirm how confounders were addressed in the analysis.Ensure that the abstract captures the novelty of the research and emphasizes how the findings can inform public health strategies.

Recommendations for Revision

1. Clarify the methodology for statistical analysis, particularly the criteria for including variables in the multivariate model.

2. Expand the discussion of significant findings, linking them to potential public health interventions or implications.

3. Address limitations and potential biases more explicitly, as well as their impact on the study's conclusions.

4. Provide details on ethical approval, participant recruitment, and informed consent to strengthen the study's credibility.

5. Improve table formatting and streamline the narrative to avoid redundancy between text and tables.

The manuscript is well-structured and provides detailed methodology. However, the narrative can benefit from clearer explanations and reduced redundancy in sections like demographics, sampling, and data collection. Ensure that all abbreviations are consistently used and explained. Strengthen the link between the qualitative data from key informants and the quantitative results, highlighting how this adds depth to the analysis.

Reviewer #2: Thanks for the opportunity to review this paper on an important public health topic. I have the following comments:

1. There is a missing word after "reduction in financial..." in line 46.

2. In line 50, I propose the authors just say children as opposed to saying "children under 5 years and over 5 years..."

3. Line 83, the authors undertook this study in "five wards" which is contradicting what is in the title and abstract

4. Line 86, there is a missing word after "in one rural health...". Is this a centre or clinic?

5. It would be interesting to know why controls were recruited from neighbors when they could have been recruited from negative individuals in the RDT register and in so doing offset the chances of misclassification of controls

6. Were the study participants all adults or the sample included both adults and children? This is not clear.

7. Under eligibility, the exclusion criteria again refer to five wards. Kindly correct this.

8. The authors also mention individuals who did not give verbal or written consent. What form of consent was approved by the IRB when the protocol was approved. It is not standard practice to have others consent verbally while others are offered written consent in the same study

9. Where were the interviews conducted? How was confidentiality assured?

10. How did the researchers ensure that all research assistants asked the questions the same way during interviews. There are high chances that interpretation of English to local language at interviewing participants might have introduced error

11. Capitalize the "d" in the title "district Medical Officer". Insert commas where appropriate is a number of sentences.

12. Line 176, clarify whether variables included in the logistic regression were significant in the univariate analysis. Is the p-value = 0.25 or <=0.25?

13. The authors should say the obtained ethical approval/permissions as opposed to "sought ...".By just seeking it is not apparent whether they got it

14. Table 1, the number of cases and controls is not consistent for the variables. Instead of 94 cases, some are 93 and some are 91 and similarly the total for controls is sometimes 90 instead of 91. There is need to ensure that the data is cleaned before analysis.

15. Table 1, under employment status, how did the authors define employment status? Housewife for example is not an employment status. A housewife could be employed or unemployed.

16. Line 209-210, the sentences should read "...were less likely to contract malaria" to make sense.

17. Table 2, include the two-by-two table for the risk factors

18. It would be important for authors to discuss why traditionally protective factors e.g. owning a treated net and sleeping under it are not significant in this population.

19. Table 3, the variable like "engages in morning and night church activities" is not there in the univariate table or have been coined differently.

20. Table 4, on the question of effectiveness of chemical to spray homes, how did the authors expect the participants to know? Were you looking for their perceptions?

21. The authors did not discuss the qualitative finding that this community resides in the fields far away from their sprayed homesteads as this is critical where the protective intervention is IRS? Is this a suitable intervention?

22. The practice of wearing long clothes that cover the whole body should be discussed in the light of the high temperatures in this area

23. Line 295, it is not clear what a cost measure is? The statement is better rephrased

Reviewer #3: Review on PloSOne Manuscript

Title: Risk Factors for Contracting Malaria in Six Wards of Mudzi District, Zimbabwe: A Case

Control-Study.

Type: Original article

General comment:

The authors performed a case-control study to identify risk factors for contracting malaria in Mudzi district, Zimbabwe. The study was well designed and reported and therefore deserves publication despite some caveats that need to be addressed. The authors identified outdoor activities in early night as a major factor contributing to malaria transmission in the study sites(The significant risk factors for contracting malaria were; engaging in night outdoor social and religious activities (AOR=8.13; 95% CI,1.74-37.90).

Specific comments:

1. The authors identified outdoor activities in early night as a major driver for transmission in Mudzi district (see in results). This finding will be more significant if supported by secondary data on entomology, for eample dating on vector biting behaviour in the study site.

2. The authors also identified Farming (gardening) activity as driver for transmission. In the discussion the authors associate this gardening activities with the “man-made” mosquito breeding sites. As “farming activities” is an important human socioaceconomic activity that support the life of the society, it is important to promote a vector unfriendly farming system, by avoiding formation of water bodies that support larval development (larval source management).

6. PLOS authors have the option to publish the peer review history of their article (what does this mean? ). If published, this will include your full peer review and any attached files.

**Do you want your identity to be public for this peer review?** For information about this choice, including consent withdrawal, please see our Privacy Policy .

Reviewer #1: No

Reviewer #2: No

Reviewer #3: No

---

## [Author Response · Author response to Decision Letter 1]

9 Apr 2025

REVIEWER #1 REVIEWER COMMENT AUTHOR COMMENTS Changed section

1. Research Gap and Study Objective (Lines 11–17):

The abstract begins with the global impact of malaria and provides context for the study's importance. However, despite the well-performing IRS, the research team highlights the persistence of malaria transmission in Mudzi. For example:

"Despite significant reductions in the malaria burden in Zimbabwe, persistent high transmission in districts like Mudzi indicates a potential role of behavioral and environmental factors unaddressed by current interventions"

o Clearly articulate how the study addresses a novel aspect or adds to existing literature. The justification has been modified to read as follows

Despite a well-performing IRS, the persistent malaria transmission in the district reveals the critical role of vectoral behavior, human behavior, and health system factors which have not been fully investigated. This case-control nested in a larger study, was conducted to determine the risk factors responsible for persistent malaria occurrence in the district

Line 17-20

2. Study Design and Methods (Lines 17–23):

o The methods are well-described, but the recruitment process for cases and controls needs clarification. Specifically:

Cases were randomly recruited from malaria facility registers, whilst controls were recruited from a village household register. A case was defined as an individual residing in a selected village who tested malaria-positive by RDT. A control was an individual with no recorded positive RDT result in the facility or community health worker malaria register

Line 23-27

Were the controls matched in any way beyond being neighbours? If not, why was an unmatched design chosen? Controls were not matched in any way beyond being neighbours. An unmatched design was used due to the likelihood of differences in characteristics of the households, including preventive, socio-economic characteristics.

Specify the criteria for classifying a participant as a "case" beyond malaria facility registers.

The use of the Kobo Collect tool is noted but lacks justification. Add a brief rationale for its use (e.g., efficiency, accuracy, or real-time data collection in rural settings).

A case was defined as an RDT-positive individual from facility registers residing in a selected village.

The justification for the use of Kobo Collect has been inserted: Kobo Collect was used for real-time data collection……. Line24-27

Lines 30-31

3. Key Findings and Interpretation (Lines 24–28):

The findings are reported with adjusted odds ratios (AOR) and confidence intervals. However:

The abstract lists the results but does not provide sufficient interpretation. For example, why might outdoor night activities (religious and social) and gardening increase risk? How do these findings align with or differ from other studies? A brief interpretation of the results has been provided in the revised abstract

Higher night outdoor activity for social and religious purposes increases the exposure of people to mosquito bites

The comparability and contrast of our study to other studies have been included in the main manuscript discussion section

Lines 38-41

Line 332-340

359-371

Briefly address why knowledge about malaria (high in both groups) did not translate to reduced risk for cases.

An explanation of why high knowledge did not translate to protective behavior is provided in the revised discussion section:

Key determinants cited in the section:

- Unemployment is pushing local communities to engage in artisanal gold mining for survival.

- Migratory farming practices

- Social needs (religion and entertainment)

- Cross-border movement for trading purposes and connecting with extended family or social connections across borders

Lines 37-47 and 372-386

4. Conclusions and Recommendations (Lines 31–33):

o The conclusion emphasizes behavior change strategies to complement IRS but lacks specific recommendations. Strengthen this by suggesting actionable measures, such as targeted educational campaigns or community-based interventions during high-risk periods (e.g., night religious gatherings).

Detailed recommendations have also been provided in the main text

A brief, compressed, actionable recommendation is given in the abstract:

Sustaining IRS and intensifying, integrated, targeted community engagement and malaria awareness programs will be key in eliminating malaria in Mudzi Line 41-44

1. Clarity and Presentation of Demographics (Lines 193–201):

The demographic results mentioned are precise but could be more concise and reader-friendly. For example, instead of stating, "There were no significant differences (p>0.05) between cases and controls on demographic variables," consider rephrasing to group insights, such as:

"Age, marital status, employment status, and income distribution were comparable between cases and controls, with no statistically significant differences observed (p>0.05)."

o Ensure the comparisons align with the data shown in Table 1 and highlight any relevant demographic findings. Thank you for this valuable suggestion.

The description of Table 1 has been modified as recommended to read:

“Case and controls were comparable on demographic variables such as age, marital status, employment status, and income distribution, with no statistically significant differences observed (p>0.05), Table 1.

Table 1

Line 237-242

2. Sampling and Study Design (Lines 122–139):

The explanation of sampling is detailed but lacks clarity in connecting the rationale for multi-stage sampling to the study objectives. Consider explicitly justifying the choice of six wards and how they adequately represent the district. The comment is well appreciated. Further clarification of how the six villages were selected has been provided.

Since the study is nested in a larger study further details have also been provided on the primary study sampling design, which informed the current study. Additional files have also been added to clarify the sampling design. Supplementary file 1 and Fig 1 has been provided to give clarity on the geospatial distribution of the selected sites.

Lines 128-141

It might help to explain why neighbors of cases were chosen as controls and how potential confounding factors, such as shared environmental risks, were accounted for.

Neighbors were chosen from the same village to allow for comparability.

Control households were within the same environment and had similar socio-behavioural aspects, but with no infected individual.

Logistical reasons and cost also informed the selection of the control from the case neighborhood

Possible known confounders, such as differences in Age, Sex, housing structure, and access to a community health worker, were accounted for in the MVLA.

3. Key Informant Interviews (Lines 157–164):

The role of the key informant interviews in complementing the case-control study is not clearly explained. What specific insights or triangulation did the interviews add to the findings? A clear link between these data and the overall research objectives would enhance the manuscript.

The comment is well appreciated.

Additional justification for Key informant interviews has been added to the text.

“Key informant interviews (KI) were also conducted to provide additional qualitative perspectives on malaria transmission drivers, primary community socio-economic behaviour, and health system capacities.:”

Line 180-189

4. Data Analysis (Lines 166–179):

The criteria for selecting variables with a p-value of 0.25 for logistic regression should be briefly justified, as this threshold may not be intuitive to all readers.

The Shapiro-Wilk test is used for normality testing, but it would help to mention whether data transformations were applied for non-normal data or how this affected subsequent analyses. No data transformations were done. Mean and standard deviations were used for normally distributed data, and median and interquartile range were used for data deviating from normality.

5. Knowledge Scoring (Lines 180–181):

The method for assessing malaria knowledge is straightforward but would benefit from additional context. How were these knowledge thresholds determined? Were they adapted from a prior study or guideline? Clarify how knowledge was assessed across the case and control groups.

Your comment is highly appreciated. We have provided additional information on how knowledge was assessed in this study.

Knowledge was assessed using the modified Bloom’s total knowledge and attitude cut-off points as shown in the revised text

A modified Bloom cut-off for total knowledge and attitude was used to assess knowledge of malaria using proportions scores methods described by Lopez et al.(42) and Okello et al. (43) where a score of 80-100% is good knowledge, 60-79 is moderate knowledge, and <59% is poor knowledge. Line 212-215

6. Ethics and Approvals (Lines 183–190):

Researchers ensured participant confidentiality during the physical follow-ups in villages, but the methods used to maintain it are not mentioned in the ethical considerations. Adding a brief note on this aspect would strengthen the ethical rigor of the manuscript. Additional information on how confidentiality was maintained during and after the study has been added to the text.

Access to the Kobo Collect storage was restricted to the researchers with usernames and passwords.

The collected information was not transferable.

After submission of each electronic questionnaire, the data was not retained on the individual phone or tablet.

Personal data was only collected on consent forms for Ethical review and compliance purposes only. The data was not included in any analysis.

Lines 223-229

Major Comments

1. Clarity and Presentation of Demographics:

The authors can improve the narrative describing the findings (lines 193–201) to enhance the demographic results' readability in Table 1. For example, the statement "There were no significant differences (p>0.05) between cases and controls on demographic variables" could be supported with more concise explanations or grouped insights, such as by stating, "Age, marital status, employment status, and income distribution were comparable between cases and controls."

This observation is highly appreciated. The necessary adjustments have been made to the description of demographics. Case and controls were comparable on demographic variables such as age, marital status, employment status, and income distribution with no statistically significant differences observed (p>0.05), Table 1. There was no significant difference between cases and controls in terms of gender, education, marital status, employment religion, and income status Line 232-237

2. Risk Factors:

The analysis of risk factors (lines 204–219) is valuable, but there is some redundancy in presenting odds ratios in both text and tables. Consider summarizing only the most significant findings in the text (e.g., wearing protective clothing, sleeping in sprayed rooms, and attending church services) while directing readers to the table for full details.

The multivariate analysis (Table 3) is compelling, but the forward stepwise method used for model building lacks detail. Explain the criteria used to include or exclude variables in the model, as this is critical to assess the robustness of the findings. The comment is well noted. A revision has been done in the manuscript to reduce redundancy

The methods section has been expanded to give more detail on how variables were included and excluded in the Multivariate model.

The p=0.25 cut-off is based on the Hosmer and Lemeshow's purposeful selection of covariates, whereby in the univariate analysis, any variable having a significant univariate test at some arbitrary level (0.25 in our case) is selected as a candidate for the multivariate analysis. Line 200-211

3. Knowledge of Malaria Transmission:

The authors could further discuss the statistical significance of differences in prevention actions (p=0.05) and perceptions of chemical efficacy (p<0.001) to explore their practical implications for public health interventions. For example, how can this information be leveraged to enhance malaria prevention strategies?

Implications of intervention and remedial actions are fully discussed in the revised text

Differences in perception of efficacy and knowledge of preventative measures within the population will result in differential uptake

Line 426-432

4. Health Service Delivery:

The section on health service delivery (lines 231–233) is underdeveloped. While it is stated that most participants felt they received good treatment, the specific measures used to assess "good treatment" or "friendly staff" are unclear. Elaborate on how these perceptions were measured (e.g., through structured interviews or surveys) and consider the potential for recall bias in self-reported data. The assessment of perceptions and attitudes toward malaria treatment services was conducted using a structured questionnaire.

The possible likelihood of recall bias occurrence in this case study is noted and acknowledged in the limitations section.

Line 25-28

5. Statistical Reporting and Interpretation:

The presentation of p-values and confidence intervals is appropriate; however, the manuscript lacks a discussion on potential confounding factors that may affect the observed associations. For example, were socioeconomic factors or healthcare access considered potential confounders? This observation is highly appreciated

The possible confounders

( age, sex, habitation status, access to community health care workers, and net ownership ) were considered in the Multi variable logistic regression analysis Line 244-249

In Table 3, the AOR for attending morning and night church services (8.13; 95% CI: 1.74–37.90) has a wide confidence interval, suggesting limited precision. This warrants further discussion regarding potential issues such as sample size or variability in this subgroup. The observation is appreciated. The observed wide confidence interval could be due to the sample size for the particular variable.

6. Ethical Considerations and Study Limitations:

There is no mention of ethical approval or participant consent in the provided text. Ensure this information is clearly stated in the manuscript.

The authors should acknowledge the study's limitations, including reliance on self-reported behaviors and potential selection bias in recruiting cases and controls. They should also discuss the generalizability of the findings to other populations or regions  The ethical statement has been provided in the text

The limitation section has been reviewed to incorporate these valuable comments. Recall bias and the likelihood of misclassification bias of cases and controls, and implications are explained in the revised limitation sections. Lines 204-217 and Lines 374-379.

Minor Comments

2. Statistical Analysis Details (Lines 23–24):

Replace "stepwise forward multivariate logistic regression analysis" with a more concise phrase, such as "multivariable logistic regression analysis." Avoid overly technical jargon in the abstract. The comment is highly appreciated.

The statement has been amended in the abstract.

Line 32

1. Terminology Consistency (Lines 15–17):

o Replace "well-performing indoor residual spraying program" with more precise language, such as "effective indoor residual spraying (IRS) program," to avoid redundancy. The statement has been revised to read “Despite an high IRS coverages in the district…” Line 17

4. Knowledge Assessment Results (Lines 28–30):

The phrase "knew at least one major method of controlling malaria" is vague. Specify what constitutes a "major method" (e.g., use of mosquito nets, IRS). The statement has been revised to read as follows:

The majority of cases (96.74%) and controls (92.22%) had good knowledge of malaria transmission and preventative measures Line 37-40

3. Protective Factors (Lines 27–28):

o The phrase "putting on long clothes that cover the whole body at night" is wordy. S

---

## [Decision Letter · Decision Letter 1]

PONE-D-24-44171R1Risk Factors for Contracting Malaria in Six Wards of Mudzi District, Zimbabwe: A Case Control-Study.PLOS ONE

Dear Dr. Fambirai,

Thank you for submitting your manuscript to PLOS ONE. After careful consideration, we feel that it has merit but does not fully meet PLOS ONE’s publication criteria as it currently stands. Therefore, we invite you to submit a revised version of the manuscript that addresses the points raised during the review process.

We look forward to receiving your revised manuscript.

Kind regards,

Rajib Chowdhury, M.Sc.; MPH

Academic Editor

PLOS ONE

Journal Requirements:

Reviewers' comments:

Reviewer's Responses to Questions

**Comments to the Author**

1. If the authors have adequately addressed your comments raised in a previous round of review and you feel that this manuscript is now acceptable for publication, you may indicate that here to bypass the “Comments to the Author” section, enter your conflict of interest statement in the “Confidential to Editor” section, and submit your "Accept" recommendation.

Reviewer #2: All comments have been addressed

Reviewer #4: (No Response)

2. Is the manuscript technically sound, and do the data support the conclusions?

Reviewer #2: Yes

Reviewer #4: Yes

3. Has the statistical analysis been performed appropriately and rigorously? 

Reviewer #2: Yes

Reviewer #4: Yes

4. Have the authors made all data underlying the findings in their manuscript fully available?

Reviewer #2: Yes

Reviewer #4: Yes

5. Is the manuscript presented in an intelligible fashion and written in standard English?

Reviewer #2: Yes

Reviewer #4: Yes

6. Review Comments to the Author

Reviewer #2: I have reviewed the manuscript as requested and I am delighted to share that I have no further comments

Reviewer #4: The residual malaria case is intriguing, particularly the approach of investigating underlying causes via a malaria risk factor study conducted through a community survey.

The study found that nighttime activities significantly contribute to malaria transmission. Farming activities were also identified as a factor that triggers the spread of malaria. However, there was limited discussion regarding mosquito entomology and behavior. Furthermore, the mechanisms of local vector transmission, whether from prior research or studies conducted in other African nations, were not clearly addressed.

Line 82 - 83: The Anopheles gambiae complex is identified as the vector transmitting malaria. What behaviors of this vector have been observed at the study site in earlier research? Additionally, what is the current status of insecticide resistance, especially regarding those used in indoor residual spraying (IRS)?

Line 84 - 88: It is mentioned that Mudzi is one of the four districts in Zimbabwe where malaria transmission still occurs despite the implementation of an IRS program (with high performance) over the past five years. It is added that despite a significant decrease in cases, malaria remains the leading cause of morbidity in the district. However, no figures are provided on the extent of the decrease in cases, the number of remaining cases, or the proportion of the malaria burden among the total population (e.g., in the API).

Line 90 - 93: This study is part of a larger project to evaluate the entomological, social, and health system determinants of malaria transmission in Mudzi. To better understand how the transmission mechanisms will be explored through questionnaires and interviews, it is necessary to explain the relationship between this study and other parts of the project, particularly those related to entomology.

Line 279 - 285: It is surprising that the questions regarding malaria knowledge did not include inquiries about local malaria vector mosquitoes, such as the abundance of mosquitoes inside and outside the house, the time and place of malaria transmission, and the habitat of Anopheles larvae in the vicinity of the house.

Line 300 - 302: The study indicates that many residents are traditional miners, which puts them at risk for malaria transmission in their work environment. Why wasn't this fact included in the malaria risk factor questionnaire?

Line 311 - 314: Many local residents participate in religious meetings and services early in the morning and at night, often leaving them unprotected from mosquito bites. Are local malaria vectors also highly active in seeking blood meals during these times?

Line 315 - 317: Communities in wards 1 and 2 practice shifting cultivation and spend a significant amount of time during the rainy season (November-April) in agricultural areas, where malaria protection is very low. Malaria cases also increase during this period (lines 78 - 80). It should be specified with figures how many malaria cases increased/decreased according to local climatic conditions.

Line 321 - 323: Cross-border migrants significantly contribute to malaria transmission, making it a critical issue to tackle. Do you have information on malaria cases linked to these migrants? If available, please include these figures in the manuscript.

Lines 328–330: If the district has implemented routine SBCC and Pre-IRS programs, why was the effectiveness of these programs not further explored in the malaria risk factor questionnaire?

7. PLOS authors have the option to publish the peer review history of their article (what does this mean? ). If published, this will include your full peer review and any attached files.

**Do you want your identity to be public for this peer review?** For information about this choice, including consent withdrawal, please see our Privacy Policy .

Reviewer #2: No

Reviewer #4: No

---

## [Author Response · Author response to Decision Letter 2]

19 Jun 2025

Line 82 - 83: The Anopheles gambiae complex is identified as the vector transmitting malaria. What behaviors of this vector have been observed at the study site in earlier research? Additionally, what is the current status of insecticide resistance, especially regarding those used in indoor residual spraying (IRS)?

Routine Insecticide Resistance, Wall cone Bioassay, and Vector Bionomics are routinely conducted in the study district and other malaria endemic districts in Zimbabwe.

Entomological surveillance reports have shown a predominant outdoor resting and outdoor biting preference for An. gambiae ss, An. arabiensis ss and An. funestus ss

Routine wall cone bioassay after Indoor residual spraying activities has shown chemical efficacy/decal rates consistent with WHO thresholds.

Resistance has not yet been identified for the pyrethroids and neonicotinoids, IRS chemicals in use.

Based on the national entomological surveys in the district, peak biting periods were established to be early evening, midnight, and early morning.

The vector behaviour, composition, and feeding and resting behaviour in the study district are well documented in the two (2) cited National Entomological Reports published in 2019 and 2022

The reports were originally available on the PMI Vector Link Websites; however, currently, the documents are not available online. Since the projects were USAID-funded, the current freeze could have affected their availability.

I have provided a link to a personal Google Drive link for the two (2) documents.

Google Drive Link: 2022 Report: https://drive.google.com/file/d/1tQ1q8M3NRnYxzZJit5tzDWu8lv_XUGZt/view?usp=sharing

2019 Report: https://drive.google.com/file/d/1UzGIGvVULBTaxxq2ZZya707C6Su5i_PL/view?usp=sharing

The background and discussion sections have been modified to capture the vital vector bionomics information from the entomological surveys.t

Line 84-Line 86

Line 84 - 88: It is mentioned that Mudzi is one of the four districts in Zimbabwe where malaria transmission still occurs despite the implementation of an IRS program (with high performance) over the past five years. It is added that despite a significant decrease in cases, malaria remains the leading cause of morbidity in the district. However, no figures are provided on the extent of the decrease in cases, the number of remaining cases, or the proportion of the malaria burden among the total population (e.g., in the API).

This comment is well appreciated

Additional information on trends in incidence, API, and IRS coverages has been added to the text. Additional Supplementary files 2 and 3 have been attached.

The revised text now reads:

Malaria burden has significantly declined in Zimbabwe since 2000; however, high-burden districts like Mudzi, located on the country’s border regions, continue to record high malaria incidence rates and Annual Parasite Index (API), ranging between 100 and 200/1000 population between 2016 and 2021(32,38,39) Supplementary File 3. Entomology surveillance has shown the absence of insecticide resistance to IRS chemicals used in the national malaria vector control program (36,40). Efficacy testing results have also been consistently within the WHO pesticide monitoring thresholds (36,40).

Line 80-Line 82

Line 95-98

Lines 90 - 93: This study is part of a larger project to evaluate the entomological, social, and health system determinants of malaria transmission in Mudzi. To better understand how the transmission mechanisms will be explored through questionnaires and interviews, it is necessary to explain the relationship between this study and other parts of the project, particularly those related to entomology.

Additional text to link this study to the sub-studies has been added to the manuscript. The text now reads as follows:

The primary factors for persistent malaria transmission in Mudzi, despite a high-performing IRS program, have not been fully explained. A broad project was conducted to evaluate the entomological, social, and health system primary factors responsible for persistent malaria transmission in Mudzi. Six sub-studies were conducted

(i) A study to identify vectoral composition and abundance in six wards.

(ii) Focus Group Discussions to identify Knowledge and Attitude, and Perceptions among community members on malaria.

(iii) Cross-sectional study incorporating Parasitaemia survey among artisanal miners and community members to identify practices, behaviour, and malaria Parasitaemia level

(iv) Cross-border movement assessment among malaria cases

(v) An assessment of the health system to establish health system capacities to deal with malaria.

The entomological studies were intended to identify the vector behaviors, whilst this nested case-control study was specifically conducted to establish the human social and behavioral determinants of malaria in six (6) wards of Mudzi District. Vector and human interaction patterns and mechanisms are critical determinants for malaria transmission.

Line 95-107

---

## [Decision Letter · Decision Letter 2]

Risk Factors for Contracting Malaria in Six Wards of Mudzi District, Zimbabwe: A Case Control-Study.

PONE-D-24-44171R2

Dear Dr. Fambirai,

We’re pleased to inform you that your manuscript has been judged scientifically suitable for publication and will be formally accepted for publication once it meets all outstanding technical requirements.

Kind regards,

Rajib Chowdhury, M.Sc.; MPH

Academic Editor

PLOS ONE

Additional Editor Comments (optional):

Reviewers' comments:

Reviewer's Responses to Questions

**Comments to the Author**

1. If the authors have adequately addressed your comments raised in a previous round of review and you feel that this manuscript is now acceptable for publication, you may indicate that here to bypass the “Comments to the Author” section, enter your conflict of interest statement in the “Confidential to Editor” section, and submit your "Accept" recommendation.

Reviewer #2: All comments have been addressed

Reviewer #4: All comments have been addressed

2. Is the manuscript technically sound, and do the data support the conclusions?

Reviewer #2: Yes

Reviewer #4: Yes

3. Has the statistical analysis been performed appropriately and rigorously? 

Reviewer #2: Yes

Reviewer #4: Yes

4. Have the authors made all data underlying the findings in their manuscript fully available?

Reviewer #2: Yes

Reviewer #4: Yes

5. Is the manuscript presented in an intelligible fashion and written in standard English?

Reviewer #2: Yes

Reviewer #4: Yes

6. Review Comments to the Author

Reviewer #2: I have no further comments. The authors have attended to all my comments. The statistical analysis is appropriate and presented well.

Reviewer #4: I have reviewed the revised manuscript.

The authors have integrated their responses into the text.

I have no further comments.

7. PLOS authors have the option to publish the peer review history of their article (what does this mean? ). If published, this will include your full peer review and any attached files.

**Do you want your identity to be public for this peer review?** For information about this choice, including consent withdrawal, please see our Privacy Policy .

Reviewer #2: **Yes: ** Addmore Chadambuka

Reviewer #4: No

---

## [Editor Report · Acceptance letter]

PONE-D-24-44171R2

PLOS ONE

Dear Dr. Fambirai,

I'm pleased to inform you that your manuscript has been deemed suitable for publication in PLOS ONE. Congratulations! Your manuscript is now being handed over to our production team.

Kind regards,

on behalf of

Dr. Rajib Chowdhury

Academic Editor

PLOS ONE